# Validation of the Computerized Cognitive Assessment Test: NNCT

**DOI:** 10.3390/ijerph191710495

**Published:** 2022-08-23

**Authors:** Itxasne Oliva, Joan Losa

**Affiliations:** Ubikare Zainketak S.L, Avenida Axpeko Erribera (pol Axpe, ed c), 11, 48950 Erandio, Spain

**Keywords:** cognition, dementia, neuropsychology, serious games, computerized

## Abstract

Population aging brings with it cognitive impairment. One of the challenges of the coming years is the early and accessible detection of cognitive impairment. Therefore, this study aims to validate a neuropsychological screening test, self-administered and in software format, called NAIHA Neuro Cognitive Test (NNCT), designed for elderly people with and without cognitive impairment. This test aims to digitize cognitive assessments to add greater accessibility than classic tests, as well as to present results in real time and reduce costs. To this end, a comparison is made with tests such as MMSE, Clock Drawing Test (CDT) and CAMCOG. For this purpose, the following statistical analyses were performed: correlations, ROC curves, and three ANOVAs. The NNCT test evaluates seven cognitive areas and shows a significant and positive correlation with other tests, at total and subareas levels. Scores are established for the detection of both mild cognitive impairment and dementia, presenting optimal sensitivity and specificity. It is concluded that the NNCT test is a valid method of detection of cognitive impairment.

## 1. Introduction

According to WHO [1], most of the population has a life expectancy of 60 years or more. Looking ahead, it is expected that by 2050 the world’s population over 60 years of age will reach 2 billion. Aging brings with it physical and cognitive changes, as well as health effects. 

At the cognitive level, some of these impairments are considered a normal part of the aging process, such as memory difficulties, reduced naming ability or deterioration of visuospatial skills. These changes may be part of healthy aging, but if this impairment of cognitive abilities is more severe, it may be mild cognitive impairment or dementia [2].

In recent years, different research has been carried out to innovate new methods for both cognitive stimulation and detection of cognitive impairment. One of these new methods is the so-called “serious games” in remote format: 

Several authors have started to use this type of format in neuropsychological assessments, such as Morrison et al. [3] who created the self-administered Revere software as an adaptation of the Rey Auditory Verbal Listening Test (RAVLT), which was validated in cognitively healthy patients with mild cognitive impairment. These authors affirm that these computerized tests are a great advance in neuropsychology. It offers very sensitive algorithms to detect possible impairments and, in the same way, to assess progress. Likewise, their results are automatically corrected, reducing the possibility of making mistakes in their correction.

Along with this, new neuropsychological assessments in serious game format have also been developed. Authors such as Zhang and Chignell [4], point out that cognitive tests based on serious games have advantages compared to classic tests performed with paper and pen. First, it is noted that the feeling of being tested is less when testing with serious game format, since paper and pen can cause anxiety or stress. Secondly, the electronic format provides greater accessibility and in this way people who live in rural areas can access this type of test. In addition, neuropsychological evaluations in serious game format reduce costs, a factor that could help people with low resources to access assessment or an intervention [5,6]. Thirdly, Mackin et al. [6] mentions that computerized neuropsychological tests allow a large amount of data to be collected much more efficiently than classical tests. These tests can record additional data such as the time spent on each answer, the time of day the test is taken or the movements of the time [7]. For this reason, they are suitable for repeated assessments and for investigating the effects of practice or familiarity with the tests, a fact that has been little studied so far. They conclude that the greatest advantage of computerized tests is the long-term follow-up of dementia [6]. 

One example of serious game format valuation is CAB (Cognitive Assessment Battery), which is composed of tests that assess the following cognitive areas: speed and attention, learning and episodic memory, visuospatial functions, language, and executive functions. It should be noted that several of the games are inspired by earlier tests such as the Boston vocabulary test [8] or the classic Stroop test [9]. The CAB is designed to take about 15–25 min [10] and can be used as an adjunct to the MMSE, and it can be a good screening test for mild cognitive impairment as it has good specificity and sensitivity. In addition, they point out that one of the strengths of the CAB is it provides a score profile that says which cognitive areas are impaired.

Among other tests that use serious computerized games is the CogState, which takes about 20 min to complete. Fredrickson et al. [11] noted that the CogState test has no ceiling effect and that it has good test-retest reliability, but at the same time they point out certain limitations such as the low validity of tasks like the card test, which has a format that is too similar to a game and far from traditional tests. In addition, this test is based solely on learning and problem solving [12]. Moreover, Mielke et al. [13], conducted a neuroimaging study comparing the results of the CogState test and classical neuropsychological tests. The results showed that CogState subtests present weak associations with markers obtained from classical tests. For their part, Hammers et al. [14], investigated with a brief version of the CogState that included four subtests assessing psychomotor reaction times, attention and memory. The aim of the research was to evaluate the ability of the computerized test to differentiate between people with Alzheimer’s-type dementia, frontotemporal dementia, Lewy body dementia and mild cognitive impairment. The authors conclude that the tool is valid as a screening test for stable dementia but is not able to differentiate between different types of dementia.

Continuing with computerized neuropsychological tests for the detection of impairment, we find the authors Doniger et al. [15], who performed the validation of the NeuroTrax computerized battery, which evaluates memory, executive functions, spatial visual perception, verbal functions, information processing and motor skills. Their research showed that NeuroTrax was useful for the detection of mild cognitive impairment. Likewise, Dwolatzky et al. [16], validated the Mindstreams battery, designed specifically for the detection of moderate cognitive impairment. They concluded that the Mindstreams neuropsychological battery, in addition to being useful for the detection of impairment at this stage, may differentiate between degrees of impairment in older adults and provide information about different cognitive profiles.

As mentioned earlier, cognitive tests based on serious games have shown great potential for differentiating cognitively healthy older adults from those with mild cognitive impairment or dementia. Due to this fact, it has even been proposed to use these tools as digital biomarkers and contribute to early detection [17]. As mentioned by Piau et al. [18], digital biomarkers are objective and quantifiable physiological and behavioral data obtained through digital devices that can be used to predict and interpret health outcomes. For example, the Braincheck test stands out for contributing to the early diagnosis of MCI. This computerized test distinguishes with high accuracy between cognitively healthy patients, MCI and patients with dementia [19]. Likewise, the virtual reality VR test also points out that such tests are favorable for the early diagnosis of MCI as well as Alzheimer’s dementia [20]. Similarly, the CogEvo tool, which focuses on orientation games and attentional processes, is useful for assessing the early stages of cognitive impairment [21]. On the other hand, the DCS tool correctly discriminates patients with dementia but its ability to differentiate those with MCI from cognitively healthy older adults is not as good [22]. In terms of cognitive areas that are assessed as digital cognitive biomarkers, tests that analyze memory and executive functions are the most sensitive and promising for the detection of MCI and dementia [23]. However, other studies show that MCI manifests itself primarily in the cognitive areas of memory, executive functions, calculation, and orientation [24]. This is the reason why several authors such as Xiao et al. [25] underline the importance of cognitive tests that aim to detect MCI assessing at least these cognitive areas.

Finally, as for electronic devices, the SMART test, which assesses attention, memory and reaction time in older adults, was validated on tablet, smartphone and computer. The study concludes that people with MCI are more comfortable using touch screens such as a tablet or smartphone [26]. For the same reason, several cognitive tests have been designed exclusively for tablet use, e.g., CAMCI or CCS [27].

Considering the reviewed bibliography, the main objective of this research is to validate a test composed of computerized serious games. 

The specific objectives are, firstly, that the test should show adequate validity. The first hypothesis is that it correlates positively with the MMSE, CDT and CAMCOG tests, both in the total score and in the subtests. 

Secondly, it is intended that the test can distinguish between cognitively healthy people with mild cognitive impairment (MCI) and Alzheimer’s dementia (AD), establishing as a second hypothesis that people with AD have lower scores on the NNCT test compared to those with MCI or who are cognitively healthy.

As a third objective, it is expected to obtain a cut-off point for both MCI and AD. As a third hypothesis, we hypothesize that participants with MCI will score higher compared to AD.

## 2. Materials and Methods

### 2.1. Procedure

In the first phase of the research, the design and computer programming of both the NAIHA (Natural and Artificial Intelligence Health Assistant)-neuro tool and the NAIHA Neuro Cognitive Test (NNCT) were conducted, designed by psychogerontology professionals and programmed by computer engineers.

Secondly, a pilot test was conducted with older adult users who were provided with a tablet and a NAIHA-neuro user. 

In a third phase of the research, the sample was recruited to proceed to perform the CAMCOG-R neuropsychological battery that includes the MMSE and CDT tests, as well as the NNCT test using tablet. 

In the last phase, the statistical analysis was carried out and conclusions were drawn from the results obtained, using the SPSS 21 programme.

In Figure 1 below we can see the complete research process:

### 2.2. Participants

A total of 147 older adults over 65 years of age took part, divided into three groups: 70 cognitively healthy, 44 with mild cognitive impairment (MCI) and 33 with Alzheimer’s dementia (AD).

For this purpose, the research team established previous criteria to be based on, taking as a reference the criteria established in the literature for each of the diagnoses. Thus, the MMSE total score, CAMCOG total score, Global Deterioration Scale (GDS) and Clock Drawing Test (CDT) verbal command mode total score were used in order, as well as the age and academic level of the subjects.

Based on these four criteria, three independent judges with extensive experience in cognitive assessment of patients with varying degrees of cognitive impairment evaluated each of the subjects, assigning them to the Healthy Older Adults, Mild Cognitive Impairment (MCI) and Alzheimer’s-type dementia groups.

### 2.3. Instruments

#### 2.3.1. Cambridge Cognitive Examination-Revised (CAMCOG-R) Scale

The Cambridge Cognitive Test (CAMCOG) [28] is a neuropsychological battery for the assessment of cognitive impairment in older adults, which includes several cognitive tests, among them the MMSE [29]. It evaluates a total of seven cognitive areas: orientation in time and place, language: comprehension and expression, recent memory and learning, attention and calculation, ideational and ideomotor praxis, abstract thinking, and visual and tactile perception. In the same way, it contemplates several levels of difficulty, minimizing the floor and ceiling effect. The CAMCOG covers all the areas necessary to carry out the operative diagnosis in DSM-5 and ICD-10, and its maximum score is 107 points. Its application time is approximately 20 min.

In its original version [28], the CAMCOG showed a sensitivity of 92% and a specificity of 96% for a cut-off point 79/80. The internal reliability (αCronbach = 0.82) and test-retest reliability (rPearson = 0.86) were also good.

#### 2.3.2. Mini-Mental State Examination (MMSE)

The MMSE [29] is a brief screening instrument to assess cognitive status. It consists of 19 items and has a maximum score of 30 points. In addition, it includes five sections that assess the following cognitive areas: orientation, immediate memory, attention and calculation, delayed recall, and language and construction. The cut-off point for this instrument is set at 24 points.

The MMSE [29] shows a sensitivity of 87% and a specificity of 82% in detecting cognitive impairment, with a test-retest value of 0.89 and an inter-rater reliability of 0.82.

#### 2.3.3. Clock Drawing Test (CDT)

Brief test used as screening for dementia in which the subject must draw a clock. This test has two conditions: to order and to copy. In the first condition, the person is asked to draw a large round clock, placing all its numbers and hands that marked ten past eleven. In the copy version, on the other hand, the person is given a drawn clock and a sheet of paper on which he/she must copy it. In this research, the condition to order has been used.

Errors in the performance of this test give us information about deficiencies in the coordination of different cognitive aspects. Its correction is made by adding the correct execution of the drawing of the dial (2 points), numbers (4 points) and hands (4 points).

Each condition has a maximum score of 10 points, the cut-off point of the copy condition being 6 points, with a sensitivity of 92.8% and a specificity of 93.48%. The most effective cut-off point for the sum of the two conditions is 15, which for a sensitivity of 94.96 produces a specificity of 90.58% [30].

#### 2.3.4. Global Deterioration Scale (GDS)

The Global Deterioration Scale [31] is a classification system designed to classify cognitive and functional capacity at different levels in cognitively healthy or cognitively impaired older adults. This scale is composed of seven different stages, from no impairment to a very severe cognitive deficit.

In this research, the GDS scale was used as a screening tool to divide participants into six groups according to their cognitive abilities.

#### 2.3.5. NAIHA Neuro Cognitive Test (NNCT)

The NNCT test is part of a tool called NAIHA-neuro designed for cognitive stimulation through serious games. Thus, the test has been designed with a total of 10 serious games. The test offers a maximum score of 35 points.

Along with the total score, it also provides a score for each cognitive area, assessing a total of seven areas: orientation, language, memory (short-term and delayed), attention and calculation, praxis, perception, and executive functions. Thus, the NNCT lasts approximately 10 min.

It should be noted that, although the format of the games is always the same, they are dynamic. That is, for example, in the “Complete the sentence” game, the sentence varies if the same user completes the test twice over time. For this purpose, it has been considered that all the sentences, images and/or stimuli to be presented have the same level of difficulty. Figure 2 shows the cognitive areas and exercises used in the NNCT:

## 3. Results

### 3.1. Convergent Validity

First, correlations were applied to study the possible convergent validity of the different subscales of the NAIHA-neuro-level test with conceptually similar variables but assessed with a different test. In this way, it is verified that the applied subscale adequately assesses the same cognitive construct. The correlations obtained for each subscale are detailed below in Table 1:

In relation to the convergent validity of the total test with the tests that study global impairment, the correlations were significant (*p* < 0.01), positive and high, as follows: CAMCOG perception (r = 0.720), with the total MMSE (r = 0.520) and with the Clock order test (r = 0.612).

### 3.2. Establishment of Judges’ Criteria

Prior to the analysis of the sensitivity of the test, we considered the possibility of establishing a criterion of judges to provide a diagnosis for each of the subjects evaluated. With the previously established criteria, three independent judges with extensive experience in cognitive evaluation of patients with varying degrees of cognitive impairment evaluated each of the subjects, assigning them to different groups. Once all the subjects were assigned to the three groups by the tests evaluated, the Fleiss Kappa was applied to check the convergence in the established diagnoses, obtaining a value of 0.86, considered as very good convergence. Three univariate analyses (ANOVA) were also applied for the total score of the MMSE, the total score of the CAMCOG, and the total score of the Clock Drawing Test, taking as the independent variable the validated inter-rater diagnosis. The results of the ANOVA tests indicated significantly (*p* < 0.001) that the established criterion was correct in demonstrating the existence of differences between the three established cognitive functioning groups: MMSE (F(2, 144) = 42.44), CAMCOG (F(2, 144) = 207.84) and Clock Drawing Test (F(2, 144) = 182.07).

When the three ANOVAs were significant, post hoc Tukey tests were applied to study whether there were differences between groups. In all cases, significant differences were observed between healthy older adults and MCI (*p* < 0.001) and AD dementia (*p* < 0.001) and between MCI and AD dementia (*p* < 0.001), with scores being lower when there was greater pathology.

### 3.3. Sensitivity Tests for the General Scale

To evaluate the sensitivity of the instrument in its global scale, ROC curves were applied to determine the sensitivity and specificity indexes. The purpose of this type of curves is to locate those subjects who meet a condition (true positives) and those who do not (true negatives); being a dichotomous analysis, the aim is to locate subjects belonging to a group who obtain a score in the test that is within the range of their group and to discard those who are assigned to a group and are outside the range of their group’s score (false negatives and false positives).

#### 3.3.1. Healthy Older Adults vs. MCI

ROC curve analyses were performed for comparison of healthy older adults, which was the sought condition (*n* = 70), versus MCI, which would be the negative condition (*n* = 44). The results were statistically significant (*p* < 0.001). The area under the curve (AUC) had a value considered to be the mean, which was 0.726 (LI = 0.633, LS = 0.820).

The optimal score to maximize sensitivity, detecting healthy older adults, and specificity (detecting those who are not healthy, MCI) was found to be 25.5. With this score, sensitivity is set at 83% and specificity at 50%. The results can be seen in Table 2:

From the sample available for comparisons between two groups (*n* = 114), estimation was performed using contingency tables to examine what percentage of subjects are above and below the cut-off point, so that those healthy above the cut-off point are true positives, those dementia below the cut-off point are true negatives, those healthy below the cut-off point are false negatives and those with dementia above the cut-off point are false positives. The difference between groups can be seen in Figure 3:

Thus, of the prognosis based on the cut-off points, 73% were correctly identified as healthy (true positive), with 27% of false positives because they actually had a diagnosis of MCI. On the other hand, of the subjects who were below the cut-off point, 75% were correctly assigned as true negative because they were AD dementia, while 25% were false negatives because, although they were healthy, they scored below the established cut-off point.

#### 3.3.2. Healthy Older Adults Vs. Dementia AD

The first of the comparisons using the ROC curve was between healthy older adults in the positive condition, as this is the condition sought (*n* = 70), versus AD dementia, which would be the negative condition (*n* = 33) as they are those who do not meet the previous condition. The results were statistically significant (*p* < 0.001). The area under the curve (AUC) had a value considered as good, which was 0.889 (LI = 0.823, LS = 0.955).

The optimal score to maximize sensitivity, detecting healthy older adults, and specificity (detecting those who are not healthy, AD dementia) was found to be 23.5 (see Table 3). With this score, sensitivity is set at 90% and specificity at 70%.

From the sample available, to make comparisons between two groups (*n* = 103), estimation was performed using contingency tables to examine what percentage of subjects are above and below the cut-off point, so that those healthy above the cut-off point are true positives, those with dementia below the cut-off point are true negatives, those healthy below the cut-off point are false negatives and those with dementia above the cut-off point are false positives (see Figure 4).

Thus, of the prognosis based on the cut-off points, 84% were correctly identified as healthy (true positive), with 16% being false positives because they had a diagnosis of AD. On the other hand, of the subjects who were below the cut-off point, 75% were correctly assigned as true negative because they were AD dementia, while 25% were false negatives because, although they were healthy, they scored below the established cut-off point.

#### 3.3.3. MCI Vs. Dementia AD

Next, ROC curve analyses were performed for comparison of MCI, which was the sought condition (*n* = 44), versus AD dementia, which would be the negative condition (*n* = 33). The results were statistically significant (*p* < 0.001). The area under the curve (AUC) had a value considered to be average, which was 0.726 (LI = 0.633, LS = 0.820).

The optimal score to maximize sensitivity, detecting MCI, and specificity (detecting those who are not MCI but diagnosed as AD dementia) was found to be 19.5 (see Table 4). With this score, sensitivity is set at 95% and specificity at 51%.

From the sample available for comparisons between two groups (*n* = 114), the estimation was performed using contingency tables to examine what percentage of subjects are above and below the cut-off point, so that those healthy above are true positives, those with dementia below the cut-off point are true negatives, those healthy below the cut-off point are false negatives and those in the dementia group above the cut-off point are false positives. This results can be see in Figure 5:

Thus, of the prognosis based on the cut-off points, 71% will be correctly identified as MCI (true positive), with 29% of false positives because they actually had a diagnosis of AD dementia. On the other hand, of the subjects who were below the cut-off point, 89% will be correctly assigned as true negative because they were AD dementia, while 11% are false negatives because, although they are MCI, they scored below the established cut-off point.

## 4. Discussion

Firstly, one of the objectives of this research was to demonstrate the validity of the NNCT test. For this purpose, the correlations of the subscales of the MMSE, CDT and CAMCOG tests were analyzed. All subscales of the NNCT test correlate significantly and positively with some of these tests, with executive functions being the subscale in which the lowest correlation has been found. In addition, the test shows very good convergence. It should be mentioned that the CAMCOG neuropsychological battery is quite comprehensive since the cut-off point for the detection of cognitive impairment considers both age and educational level. That the NNCT test, although shorter than the CAMCOG, has achieved good correlations is very revealing. Other computerized tests such as the CogState [11] show low correlations with neuropsychological batteries that analyze different cognitive areas.

Secondly, the aim was to distinguish healthy subjects, subjects with MCI and subjects with AD. It can be affirmed that the established objective has been fulfilled since the test discriminates correctly between these three groups. In addition, those participants with AD were expected to score lower compared to MCI or cognitively healthy individuals on the NNCT test. It is considered of great value that the test may perform a correct discrimination of cognitive profiles since the NNCT can be used for diagnostic support, as well as a screening test for professional decision making in relation to the intervention plan. This study supports what was mentioned by Mielke et al. [13] that computerized tests are highly effective for neuropsychological assessment since they are created with very accurate algorithms at the detection level, thus limiting the possible errors that the professional can make in its correction. As well as what was noted by Ye et al. [19] that computer-based tests help in the early diagnosis of MCI. As previously mentioned, for a correct detection of MCI it is important to analyze the cognitive areas of memory, executive functions [23,24], calculation and orientation [24]. That is why the NNCT test performs an analysis of each of these areas. Other tests such as CogEvo [21], although suitable for the detection of early stages of cognitive impairment, focus only on the analysis of the cognitive areas of orientation and attentional processes. As the detection of MCI is complex, not all tools are suitable for it, with some tools such as DCS [22] proving only suitable for the detection of dementia.

Thirdly, the aim was to establish a cut-off score to simplify the future categorization of users of the NAIHA tool and for the detection the MCI and dementia. A score of 23.5 points was established for the detection of cognitive impairment, 19.5 points for AD and 25.5 points for MCI. All three scores have optimal sensitivity and specificity. The sensitivity of the NNCT to distinguish cognitively healthy people from those with MCI is 83%. The sensitivity of this test is higher than that shown by tests such as CogEvo (81.8%) [21] or CAB (46%) [8] for the detection of MCI. It could be argued that the NNCT test is of great value for the detection of MCI and could even be used as a digital biomarker.

Among the advantages of the NNCT test is that it evaluates a total of seven cognitive areas in a short period of time, reducing travel and waiting time for professionals since the person can self-administer the test from home. Moreover, this type of testing reduces costs and increases access for many people to know their cognitive health [5,6]. Another advantage is that it is a remote test that provides results in real time. As for professionals, the NNCT test can provide support as it allows monitoring of the patient’s cognitive status from the patient’s home. As already mentioned, people with cognitive impairment tend to feel more comfortable using touch screens [26], which are more intuitive. Arguably, this is another advantage of the NNCT, which is designed for tablets, although its use can be extended to computers or smartphones.

This is of great relevance to cognitive health as the state of older adults can change very quickly. Therefore, the NNCT test can be used to support the monitoring of the patient’s cognitive status at home. As already mentioned, this is one of the great advantages of the NNCT [6]. However, with this test we wanted to go further and developed dynamic games. This allows the person to repeat the test several times without the need to repeat some of the evaluation games, which happens in other classic or computerized tests and is a problem when monitoring the cognitive state of the patient. Therefore, it is concluded that the NNCT can be of great value in this sense because it reduces the possibility of learning the answers.

Regarding the limitations of the test, it should be mentioned that some of the participants could not complete all phases of the study because they needed good vision and ability to use tablet devices, which tends to deteriorate in advanced stages of dementia.

In future research it would be advisable to analyze each game of the test individually, as well as to expand the sample in the more advanced dementia group.

## 5. Conclusions

In conclusion, it can be stated that the NNCT test is a good screening method since both its total score and the score of the different subscales show significant correlations with previously validated and effective scales for the detection of cognitive impairment and dementia.

Additionally, NNCT correctly categorizes the users and differentiates those with dementia from those with MCI. Also, it can assist in the early diagnosis of cognitive impairment by facilitating access to cognitive assessments.

The NNCT is a breakthrough in the field of cognitive assessment remotely since, as we have seen, there are not many tests in “serious games” and computerized format that exist today to correctly detect MCI.

## Figures and Tables

**Figure 1 ijerph-19-10495-f001:**
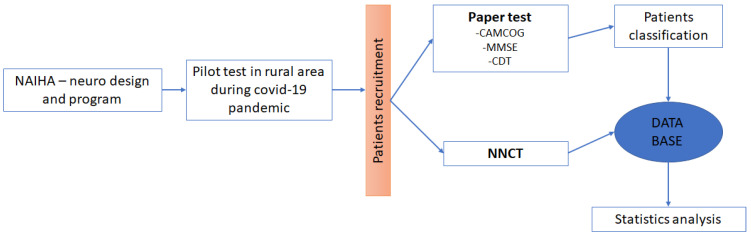
Sequence of the procedure. In this image, we can see the sequence of the complete project, from the design of the programme to the analysis of the results.

**Figure 2 ijerph-19-10495-f002:**
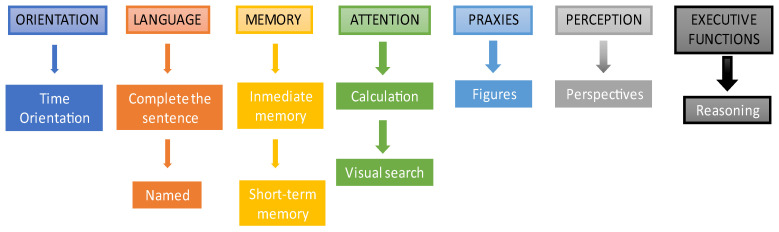
Cognitive areas and exercises of the NNCT test. The picture shows the exercises associated with each cognitive area in the order in which they are presented in the test.

**Figure 3 ijerph-19-10495-f003:**
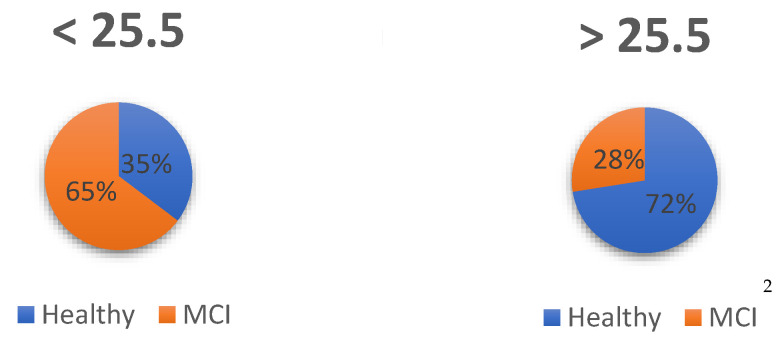
Assignment of subjects: Healthy and MCI. ^2^ The assignment of subjects was done according to diagnosis and cut-off point.

**Figure 4 ijerph-19-10495-f004:**
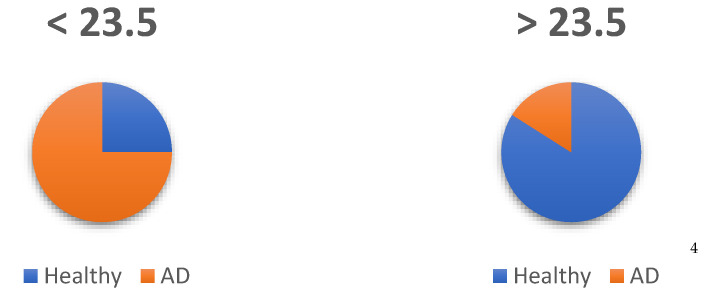
Assignment of subjects: Healthy and AD. ^4^ Number of subjects^.^ According to diagnosis and cut-off point.

**Figure 5 ijerph-19-10495-f005:**
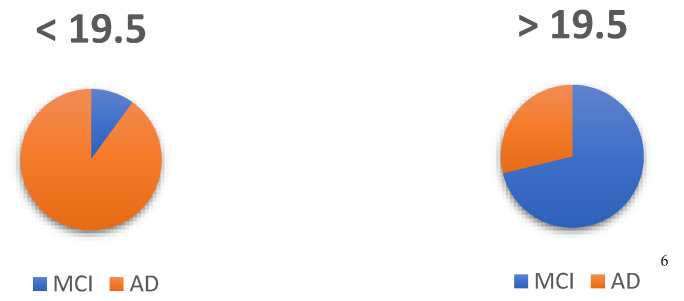
Assignment of subjects: MCI and AD. ^6^ Number of subjects^.^ According to diagnosis and cut-off point.

**Table 1 ijerph-19-10495-t001:** Correlation of NNCT subscales.

Subscales	Correlation (*p*<)	Subscale MMSE	Subscale CAMCOG	Total MMSE	CDT
Orientation	0.001	0.735	0.732	0.464	0.582
Language	0.001	0.220	0.312	0.253	0.224
Memory	0.001	0.263	0.401	0.275	0.328
Attention	0.001		0.401	0.307	0.304
Praxis	0.001		0.240	0.183	0.276
Executive Function	0.05				0.174 ^1^

^1^ Correlation between NNCT and MMSE, CAMCOG and CDT condition to order.

**Table 2 ijerph-19-10495-t002:** Healthy older adults and MCI.

NNCT Scores	Sensibility	Specificity
22.5	0.943	0.159
23.5	0.9	0.25
24.5	0.857	0.341
**25.5**	**0.829**	**0.5**
26.5	0.743	0.568
27.25	0.643	0.659
27.75	0.629	0.659
28.5	0.571	0.795
29.5	0.486	0.864
30.5	0.443	0.8861 ^1^

^1^ Sensitivity and specificity of the NAIHA-neuro global scale for discriminating between healthy older adults and MCI. Optimal cut-off point is highlighted in bold.

**Table 3 ijerph-19-10495-t003:** Healthy older adults and AD dementia.

NNCT Scores	Sensibility	Specificity
20.5	0.971	0.485
21.5	0.957	0.515
22.5	0.943	0.576
**23.5**	**0.9**	**0.636**
24.5	0.857	0.727
25.5	0.829	0.788
26.5	0.743	0.879
27.3	0.643	0.879
27.8	0,629	0.879
28.5	0.571	0.9093 ^3^

^3^ Sensitivity and specificity of the NAIHA-neuro global scale for discriminating between healthy older adults and AD dementia. Optimal cut-off point is highlighted in bold.

**Table 4 ijerph-19-10495-t004:** MCI and AD.

NNCT Scores	Sensibility	Specificity
15.5	0.977	0.303
16.5	0.977	0.333
17.5	0.977	0.364
18.5	0.955	0.364
**19.5**	**0.955**	**0.485**
20.5	0.909	0.485
21.5	0.909	0.515
22.5	0.841	0.576
23.5	0.75	0.636
24.5	0.659	0.7275 ^5^

^5^ Sensitivity and specificity of the NAIHA-neuro global scale for discriminating between healthy older adults and MCI. Optimal cut-off point is highlighted in bold.

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
