# Peer review of "Validation of the Computerized Cognitive Assessment Test: NNCT"

_ijerph, 2022, doi:10.3390/ijerph191710495_

Round 1
Reviewer 1 Report
Oliva and Losa's manuscript entitled "Validation of the computerized cognitive assessment test: NNCT" is an interesting article, where the authors evaluated 7 -cognitive areas and showed a significant and positive correlation with other tests including total and subareas levels. The authors went further to assign the score for the detection of both mild cognitive impairment and dementia, presenting optimal sensitivity and specificity. The authors concluded that the NNCT test is a valid method of detection of cognitive impairment and is highly relevant to people living in rural areas.
The strength of the article is that the MNCT test for detecting cognitive impairment is of public health interest in rural areas. However, I still have some issues below that need to be properly addressed.
1. Please provide your current affiliation.
2. Line 11:If NAIHA is used for the first time here, please provide the full form. The same applies to Line 13 for MMSE, CAMCOG, etc.
3. Line 8-17: Abstract section should mention precisely whether this study is designed for animal models, primate models, or human subjects.
4. The abstract did not clearly mention what statistics were used for making comparisons, the sex factor of the subject, and age group is not clear.
The abstract could be more clear if you can reiterate lines 51 and 52 texts here as well.
Introduction section:
Line 61: please check this sentence for grammatically error-free.
Line 65: "Moreover, [8], conducted......". I have difficulty understanding such type of writing, here and elsewhere please provide the author/author's name together with this reference. Providing the author's name along with this reference number will make the manuscript more readable.
Line 68 and elsewhere: See above in the comment section of Line # 65, how to address those texts.
Method section:
Line 95 to Line 175: Can you provide the schematic diagram/sketch of how these studies are designed? If you think the flow chart will provide better reflection, you are welcome to summarize the workflow in the figure form as well.
If you have used the scientific machines, please provide the details of those machines used (company name, model, capacity, etc).
Results section:
The opening sentence of the result should have a smooth transition from the context above. Please make sure that the transition is appropriate. Not too much jumping from the context and maintaining coherent logical flows is critical.
Line 183- Line 202: It will be more presentable if you group all these data in a Tabular form rather than explaining them in the text.
Table 1: Line 243-244: Please make sure whether the values are separated by decimal or comma? Please explore the possibility of whether you can substitute the Table entirely with the Heat Map form representation.
Table 2: I suggest presenting these data in the form of a pie chart if possible.
Line 263-264: Here and elsewhere, please do not start with the "." decimal value. Place "0" in front of the decimal.
Table 3: If possible provide the data in Heat Map. I believe the data should not use commas here and elsewhere and the use of decimals is necessary.
Tables 4 and 6: Follow comments here as mentioned in Table 2.
Table 5: Follow comments similar to Table 3.
Discussion: In the discussion section, the authors MUST compare their study against available literature, report data, or work from other studies. They should provide their opinion or reasoning why such consistency and discrepancies are happening.
- Please provide the schematic sketch incorporating summary of how your findings are relevant to the brain signaling cascade for the reader to have a bird's eye view of projection patterns in the brain and the emergence of relevant cognitive function.
The article should also mention the limitation of the study.
Conclusions: Line 334: The conclusion section should be started in a separate paragraph.
Overall, the article at the present state needs to work more on presentation style replacing most of the tabular data with figure plots. Logical flow and explanation of the statistical test used with sample number and p-values should be also taken care of. The author should also be able to highlight what are the novel findings that can move the field forward from this study.
Author Response
Thank you very much for your corrections. They are very concrete and helpful.
I have tried to correct all the points mentioned including the heat maps and graphs mentioned, as well as, an outline of the procedure. I think this simplifies and helps in the understanding of the article.
At the same time, I have included the conclusions section and I have improved the wording.
Any other contribution to the improvement of the article will be welcome.

Reviewer 2 Report
Paper should be completely rewritten as it has lots of misprints and mistakes. Moreover there are two main concerns that must be considered:
- The format is really confusing, specially with the citations. What system are you using? In the first paragraph there are two different citations format= This is unacceptable and seems that the paper has been written in a quickly way without paying the sufficient attention to this matter.
- Moreover, section relative to methods (section 2) need to be written in a really deeper way, as many readers may not be familiar with all these terms.
- There is no conclusions section. In good journals as this is, conclusions section is really important and without it, paper cannot be taken serious enough to be considered.
- There is only 15 references and many of them are too old to be considered as relevant in area, I mean you need to make a deep bibliography revision as you need to have a strong theoretical background.
Author Response
1- The format is really confusing, specially with the citations. What system are you using? In the first paragraph there are two different citations format= This is unacceptable and seems that the paper has been written in a quickly way without paying the sufficient attention to this matter.
Response 1: He modificado el formato de citaciones, ordenando alfabeticamente en la parte de bibliografia y mencionando a los autores solamente cuando es imprescindible
2- Moreover, section relative to methods (section 2) need to be written in a really deeper way, as many readers may not be familiar with all these terms.
Response 2: An scheme has been included to facilitate the understanding of the study procedure.
3 - There is no conclusions section. In good journals as this is, conclusions section is really important and without it, paper cannot be taken serious enough to be considered.
Response 3:the conclusions section has been included
4- There is only 15 references and many of them are too old to be considered as relevant in area, I mean you need to make a deep bibliography revision as you need to have a strong theoretical background.
Response 4: new biography has been added
I hope that the understanding and quality of the article has improved. Thank you very much for your corrections

Round 2
Reviewer 1 Report
The article has been immensely improved.
However, still there needs further improvement.
1. Affiliation of author’s not complying
with Journal’s guideline. It should have street adress, City, State, Country, zip code etc.
2. Discussion section still lacks adequate
references from other’s work
3. I see that numerical values are separated somewhere by comma and somewhere by decimal, please follow standard method of reporting?
4. The article still lacks a good original image to show
which brain area’s dysfunction correlates to
cognitive impairment. A MRI or similar type of
traces could be very valuable to
interpret your study.
Otherthan above, I do not have further comments.
Author Response
1. Affiliation of author’s not complying
with Journal’s guideline. It should have street adress, City, State, Country, zip code etc.
1. Corrected
2. Discussion section still lacks adequate
references from other’s work
2. The number of references has been expanded.
3. I see that numerical values are separated somewhere by comma and somewhere by decimal, please follow standard method of reporting?
3. Corrected
4. The article still lacks a good original image to show
which brain area’s dysfunction correlates to
cognitive impairment. A MRI or similar type of
traces could be very valuable to
interpret your study.
4. Since we have not used MRI, an image has been included with the cognitive areas that are assessed and the exercises that are used.
thank you very much for your contributions, they have been a great help.

Reviewer 2 Report
Theoretical framework is still insufficient, please make a deep bibliography search to make an important background that allows you to later conclude. Moreover, the first answer to reviewer is in Spanish.
Author Response
1. Theoretical framework is still insufficient, please make a deep bibliography search to make an important background that allows you to later conclude.
1. The number of references has been increased. Also, the introduction and subsequent discussion has been improved.
Thank you very much for your contributions.
